# The *in vivo* RNA structurome of the malaria parasite *Plasmodium falciparum*, a protozoan with an A/U-rich transcriptome

Franck Dumetz[1☉¤], Anton J. Enright[1☉], Jieyu Zhao[2], Chun Kit Kwok[2,3]*, Catherine J. Merrick[1]*

**1** Department of Pathology, University of Cambridge, Cambridge, United Kingdom, **2** Department of Chemistry and State Key Laboratory of Marine Pollution, City University of Hong Kong, Kowloon Tong, Hong Kong SAR, China, **3** Shenzhen Research Institute of City University of Hong Kong, Shenzhen, China

☉ These authors contributed equally to this work.
¤ Current address: Institute for Genome Sciences, University of Maryland School of Medicine, Baltimore, Maryland, United States of America
* ckkwok42@cityu.edu.hk (CKK); cjm48@cam.ac.uk (CJM)

**Data Availability Statement:** Raw sequence data (FASTQ) available from the European Nucleotide Archive (ENA) (accession number PRJEB44384).

## Abstract

*Plasmodium falciparum*, a protozoan parasite and causative agent of human malaria, has one of the most A/T-biased genomes sequenced to date. This may give the genome and the transcriptome unusual structural features. Recent progress in sequencing techniques has made it possible to study the secondary structures of RNA molecules at the transcriptomic level. Thus, in this study we produced the *in vivo* RNA structurome of a protozoan parasite with a highly A/U-biased transcriptome. We showed that it is possible to probe the secondary structures of *P. falciparum* RNA molecules *in vivo* using two different chemical probes, and obtained structures for more than half of all transcripts in the transcriptome. These showed greater stability (lower free energy) than the same structures modelled *in silico*, and structural features appeared to influence translation efficiency and RNA decay. Finally, we compared the *P. falciparum* RNA structurome with the predicted RNA structurome of an A/U-balanced species, *P. knowlesi*, finding a bias towards lower overall transcript stability and more hairpins and multi-stem loops in *P. falciparum*.

This unusual protozoan RNA structurome will provide a basis for similar studies in other protozoans and also in other unusual genomes.

## Introduction

RNA molecules, composed of single strands of four different ribonucleotides, do not adopt a single canonical structure like the double helix of DNA; instead they can fold into complex secondary structures made of loops, hairpins and bulges [1]. They serve a variety of roles in the cell [2, 3], including structural roles carried out by transfer and ribosomal RNAs (tRNAs, rRNAs) and protein coding roles for messenger RNAs (mRNAs). These roles are facilitated by the ability of RNA molecules to adopt, and transition between, highly specific secondary structures [4, 5]. Different secondary structures can impact on the location of the RNA molecules,

All other relevant data are within the paper and its Supporting information files.

**Funding:** To CJM: UK Medical Research Council (grants MR/K000535/1 and MR/L008823/1) To CKK: Royal Society Kan Tong Po fellowship, Shenzhen Basic Research Project (JCYJ20180507181642811), Research Grants Council of the Hong Kong SAR China Projects (CityU 11101519, CityU 11100218, N_CityU110/17, CityU 21302317), Croucher Foundation (Project No. 9509003), State Key Laboratory of Marine Pollution Director Discretionary Fund, City University of Hong Kong (projects 7005503, 9680261, 9667222). The funders had no role in study design, data collection and analysis, decision to publish, or preparation of the manuscript.

**Competing interests:** The authors have declared that no competing interests exist.

their metabolism and stability [6], their interaction with RNA binding proteins [7], their function and their regulation of protein expression. Two outstanding examples of this are the function-related shape of tRNA [8] and the translation expression controlling element, the Riboswitch [9].

One of the common strategies to investigate RNA structures *in vivo* is Selective 2′-Hydroxyl Acylation analyzed by Primer Extension (SHAPE) [10]. SHAPE uses chemicals that modify the 2' hydroxyl group of flexible nucleotides, in which the 2' O-adduct will inhibit reverse transcription during cDNA synthesis [11]. The pattern of stalling is then detected on a sequencing gel, which determines the identity of all unpaired/flexible bases in the RNA, and hence allows its secondary structure to be inferred. The technicalities of SHAPE only permit the analysis of one transcript at the time, requiring a high amount of starting RNA and potentially more than one sequencing gel to analyse larger transcripts. All these technical limitations rendered impossible the transcriptome-wide study of RNA structures, which is referred to as a structurome [12]. However, the advent of Next Generation Sequencing (NGS) and bio-informatics revolutionised the field of RNA structure and within the past decade, Structure-seq [13], and many other related methods [14–16], were developed.

To date, only a few *in vivo* structuromes are available. Many viruses have been studied, probably due to their small number of RNA molecules, such as HIV [17, 18], DENV and ZIKV [19, 20], influenza [21] and more recently SARS-CoV-2 [22–27]. Regarding eukaryotes, structuromes were established for model organisms like the plants *Arabidopsis thaliana* [13] and *Oryza sativa* (rice) [28], the yeast *Saccharomyces cerevisiae* [16], the bacteria *Escherichia coli* [29] and amongst mammals, mouse embryonic stem cells [29, 30] and human cell lines [6, 16, 29, 31]. These have revealed fundamental conserved features across the landscape of RNA structures.

In this paper we have established the *in vivo* structurome of a protozoan, *Plasmodium falciparum*. *P. falciparum* is one of the etiological agents of malaria, the most widely lethal human parasitic disease in the world [32]. Little is known about RNA structures in *P. falciparum*, yet the parasites offer many opportunities to understand the roles of RNA structures in eukaryotic cells. Firstly, the natural life cycle of the parasite oscillates from a warm-blooded host to an insect vector, meaning that the same organism is exposed to very different temperatures and metabolic environments. This allows the study in natural conditions of the effect of the environment and temperature upon the RNA structurome. Indeed, it has been shown that heat stress can affect the structuromes of plants [33] and bacteria [34, 35], which can likewise experience temperature extremes in nature. Furthermore, one intriguing piece of evidence concerning rRNAs suggests that temperature-responsive RNA structures are also important in *Plasmodium* life cycles. *Plasmodium* does not encode rRNAs in conventional tandem arrays, but rather in several isolated units which are transcribed independently in either mosquito stages (S-type) or mammalian stages (A-type) [36, 37]. rRNA transcription apparently switches in response to differential temperature and nutrient availability in the two hosts [38, 39]. The two types of rRNA were predicted, based on their sequences, to have different structures [40] and they do perform differently when complemented into *S. cerevisiae* [41], so there is clear potential for them to influence translation in insects versus humans. Thus, *Plasmodium* may have evolved variant RNA structures as a key element in its life cycle transitions.

Secondly, while all the previously cited organisms have a more balanced ratio of A/T and G/C in their genomes, *P. falciparum* is 80.6% A/T-rich [42]: one of the most biased genome compositions ever to be sequenced. This offers a critical point of view when studying genome evolution, and it may well lead to unusual RNA structures.

This RNA structurome of *P. falciparum* sets out intends to investigate the relationship between RNA shapes and genome composition, as well as transcription efficiency, in the asexual erythrocytic cycle of this important human parasite.

## Material and methods

### Parasite culture

*Plasmodium falciparum* 3D7, obtained from the Malaria Research and Reference Reagent Resource Center (MR4), was cultured in O+ human erythrocytes at 4% haematocrit in RPMI 1640 supplemented with 0.25% albumax (Invitrogen), 5% heat-inactivated human serum and 0.25% sodium bicarbonate in gassed chambers at 1% $O_2$, 3% $CO_2$, and 96% $N_2$ [43]. Parasite count was assessed on thin blood smears stained with Hemacolor (Merck).

### RNA structure probing and poly(A) enrichment

An asynchronous culture was grown up to $1.2 \times 10^{10}$ parasites as previously described [43]. Erythrocytes were pelleted and then incubated for 30 minutes either with 10 mM DMS (dimethyl sulfate, Sigma) in ethanol or 20 mM NAI (2-methylnicotinic acid imidazolide) in anhydrous DMSO, at 37˚C, agitated at 200 rpm, in the same gas mixture described above. The NAI was prepared as previously described [44]. The reaction was stopped with 40 mM dithio-threitol (DTT). Parasites were extracted by erythrocyte lysis using 1v of 0.2% saponin in PBS and then washed 3 times in ice-cold PBS. RNA was extracted using Qiagen RNeasy plus mini kit (Qiagen) following manufacturer's instructions and then submitted to DNase treatment for 15 minutes (Qiagen). The total RNA recovered was split into samples of 5 μg each and enriched in poly(A) transcripts using NEBNext Poly(A) mRNA magnetic isolation module (NEB) according to manufacturer's instructions.

### Visualisation of 5.8S rRNA structure using RT-stalling

RT stalling was performed using a primer targeting the 5.8S rRNA gene (PF3D7_0531800). Briefly, 1.5 μg of total RNA from DMS, NAI and DMSO-treated cultures in 5.5 μl nuclease-free water was mixed with 1 μl of 5 μM Pfa_5.8S primer labelled at the 5' end with Cy5 [Cy5] `ATTTTCTGTAGGAGTACCACT` (Eurofins Genomics) in 3 μl of reverse transcription buffer (20 mM Tris, pH 7.5, 4 mM $MgCl_2$, 1 mM DTT, 0.5 mM dNTPs, 150 mM LiCl in final). For the sequencing reactions (A/G/C/T), an extra 1 μl of 10 mM dideoxynucleoside triphosphate (Roche) was added to replace 1 μl of nuclease-free water. All samples were heated at 75˚C for 3 min, 35˚C for 5 min and then held at 50˚C. Next, 0.5 μl of Superscript III (200U/μl) (Thermo Scientific) was added and reverse transcription was performed at 50˚C for 15 min, followed by addition of 0.5 μl of 2M NaOH at 95˚C for 10 min to degrade the RNA template. After reverse transcription, 10 μl of 2x formamide orange G dye (94% deionized formamide, 20 mM Tris pH 7.5, 20 mM EDTA, orange G dye) was added and heated at 95˚C for 3 min before loading into a pre-heated (at 90W for 45 min) 8% denaturing polyacrylamide gel. The gel was run at constant power of 90 W for 90 min after loading 3.2 μl of each sample. Fujifilm FLA 9000 was used to scan the gel.

### Library preparation and sequencing details

Random fragmentation of polyA-enriched RNA was performed with 150 ng polyA-enriched RNA in fragmentation buffer (40 mM Tris-HCl pH 8.2, 100 mM LiCl, 30 mM MgCl2) at 95˚C for 60s to generate average fragment size of ~250 nt, and then purified with RNA Clean & Concentrator (Zymo Research). 3' dephosphorylation reactions included 7 μl fragmented sample, 1 μl of 10x PNK buffer (NEB), 1 μl of rSAP enzyme (NEB) and 1 μl of PNK enzyme (NEB), carried out at 37˚C for 30 min. 3' adapter ligation was performed by adding 3' adapter (5'-/ 5rApp/NNNNNNAGATCGGAAGAGCACACGTCTG/3SpC3/-3' with 1:5 molar ratio of RNA to 3' adapter), PEG 8000 (17.5% final) and T4 RNA ligase 2 (NEB) in 1x T4 RNA ligase

buffer at 25°C for 1 h. Excess adapter was digested by adding 1 µl RecJf (NEB) and 1 µl 5'dead-enylase (NEB) at 30°C for 30 min and removed by RNA Clean & Concentrator (Zymo Research). Reverse transcription reactions including the ligated RNA above, reverse primer (5'-CAGACGTGTGCTCTTCCGATCT-3' with 1:2.5 molar ratio of RNA to reverse primer), reverse transcription buffer (see RTS method above) and Superscript III (1U/µl final) (Thermo Scientific) were performed at 75°C for 3 min, 35°C for 5 min, then 50°C for 50 min. In this step, Superscript III should be added before the 50°C and after reverse transcription, NaOH (0.1 M final) was added at 95 °C for 10 min for RNA degradation, followed by mixing with 5 µl 1M Tris-HCl (pH 7.5) before RNA Clean & Concentrator (Zymo Research). The purified cDNA was then ligated with 5' adapter (5'/5Phos/AGATCGGAAGAGCGTCGTGTAGCTC TTCCGATCTN10/3SpC3/-3' with 1:20 molar ratio of RNA to 5'adapter) using Quick Ligation Kit (NEB) at 37°C overnight. Ligated reaction mixture was heated to 95°C for 10 min for inactivation and mixed with 1 volume of 2x formamide orange G dye before purifying with 10% denaturing urea-TBE acrylamide gel (Thermo Scientific) at 300V for 20 min. The size of 100–400 nt was sliced, crushed and soaked in 1x TEN 250 buffer (1x TE pH 7.4, 0.25 M NaCl) with incubation at 80°C for 30 min with 1300 rpm shaking, and then purified with RNA Clean & Concentrator (Zymo Research). The purified ssDNA was mixed with forward primer (5'-AATGATACGGCGACCACCGAGATCTACACTCTTTCCCTACACGACGCTCTTCCGATCT-3', 0.5 µM final) and reverse primer with different indexes (5'-CAAGCAGAAGACGGCATACGA-GAT-(6 nt index seq)-GTGACTGGAGTTCAGACGTGTGCTCTTCCGATCT-3', 0.5 µM final) for PCR amplification by using KAPA HiFi HotStart ReadyMix (KAPA Biosystems). The PCR program included 95°C for 3 min, 16 cycles of 3 steps (98°C: 20s, 68°C: 15s, 72°C: 40s), 72°C for 90s, then cooled to 4°C for size selection. PCR samples were resolved on a 1.8% TAE agarose gel at 120 V for 55 min. The size of 150–400 nt was cut and recovered by Zymo DNA agarose gel extraction kit (Zymo Research). DNA libraries were quantified, pooled and sequenced on the Illumina Hiseq System in 150 bp paired-end (PE) configuration. A more detail protocol was described in reference [45].

## Structurome analysis

Forward and reverse reads from each sample were cleaned of adapter sequence contamination using *reaper* (v15-065) from the *kraken* package [46]. Paired reads were then aligned to an indexed reference transcriptome (PlasmoDB-45_Pfalciparum3D7) using *Bowtie2* (v2.4.2) in paired-end mode and converted to SAM and sorted BAM files [47]. Reverse transcription stops were assessed using *StructureFold2* for each individual sample [48]. Coverage and over-lap of RT stop data were then computed along with cross-replicate coverage and overlap and stop correlation was assessed among all samples via *StructureFold2*. These data were normal-ised and structural reactivity data generated for each transcript with enough supporting data across samples using *StructureFold2*. These reactivities were supplied to *RNAStructure* for experimental reactivity constrained folding [49]. Resulting structures were then assessed for structural features using *Forgi* from the *ViennaRNA* package [50]. Unconstrained folding of every transcript was also performed using RNAStructure with the same parameters and assessed similarly to generate a cohort of unconstrained purely *in silico* folded transcripts. Parameters for the number of reads and mapping efficiency for each library are shown (Table 1).

## Functional dataset analysis

Microarray datasets from parasites exposed to hyperoxia were extracted from [51], and the chloroquine-exposed dataset was extracted from [52]. Lists of genes were updated to the

**Table 1. Sequencing parameters for each NGS library generated in this study.**

| Sample | Number of Read Pairs | Yield (Gigabases) | Mapped Reads | Mapped Reads (%) |
|---|---|---|---|---|
| Total-RNA-DMS-3 | 148,865,508 | 44.6597 | 108,912,838 | 73.1% |
| Total-RNA-DMSO-3 | 163,681,632 | 49.1045 | 115,215,035 | 70.3% |
| Total-RNA-NAI-3 | 192,424,134 | 57.7272 | 144,128,728 | 74.9% |
| Total-RNA-DMS-2 | 166,742,223 | 50.0227 | 117,106,285 | 70.2% |
| Total-RNA-DMSO-2 | 186,503,272 | 55.951 | 128,887,798 | 69.1% |
| Total-RNA-NAI-2 | 145,601,085 | 43.6803 | 108,451,225 | 74.5% |

current genome annotation. Ribosome profiling data were obtained from [53] and translational efficiency was calculated as ribosome density per messenger RNA. The RNA decay dataset was extracted from [54] at 10hpi for rings, 24hpi for early trophozoites, 30hpi for late trophozoites and 46hpi for schizonts. Structural parameters per transcript (numbers of nucleotides involved in stems, hairpins, multi-stem loop and bulges) were normalized to transcript length before comparing these structural parameters with the translational efficiency of each gene.

## Results

### Assessment of NAI and DMS as RNA structure probing chemicals for *P. falciparum* inside the erythrocyte host cell

We conducted *in vivo* structure-seq on *Plasmodium falciparum*: an organism with a >80% AT-rich genome, and one that lives for the majority of its lifecycle inside another host cell. It was necessary first to test and optimise the parameters for *in vivo* RNA probing in this unusual organism. We used two different probing agents with different nucleotide reactivity: NAI (2-methylnicotinic acid imidazolide) and DMS (dimethyl sulfate). NAI reacts with all four unpaired nucleotides while DMS selectively reacts with unpaired adenosine and cytosine. Neither of these chemicals had been used in *P. falciparum* before. In order to check if the chemicals could penetrate inside the parasite inside the host erythrocyte, we exposed an asynchronous culture of *P. falciparum* to NAI or DMS. After RNA extraction we performed a primer extension with a Cy5-conjugated primer targeting the 5.8S rRNA transcript. When a nucleotide has been modified by one of the chemicals, meaning that the nucleotide was unpaired in the original RNA, the reverse transcriptase will stall one nucleotide beforehand, stopping the elongation reaction (Fig 1A). This can be seen on a denaturing polyacrylamide sequencing gel (Fig 1B, S1 Raw images). Reading the sequence confirmed that it was identical to the PF3D7_0531800 gene sequence available on PlasmoDB [55]. In NAI and DMS probed parasites, base reactivity was seen with all four nucleotides after NAI-probing, and with only adenosine and cytosine after DMS-probing. (Modification of adenosine and cytosine was not, however, identical at all positions with DMS and NAI, because these chemical probes react with different moieties of the RNA nucleotides, and with different efficiencies.) This indicated that both chemicals could penetrate into *P. falciparum*, and our probing strategy could investigate the RNA structures in *P. falciparum*.

To prepare the structurome, we then extracted the total RNA from two independent asynchronous parasite cultures treated with NAI, DMS, or DMSO only, as in Fig 1, and used poly (T) magnetic beads to enrich poly(A) RNA. The enriched poly(A) fraction represented 2–3.4% of the total RNA, which is commensurate with the literature for eukaryotic cells [56]. The enriched fractions were then used to generate NGS libraries (Fig 2A). The sequencing outputs

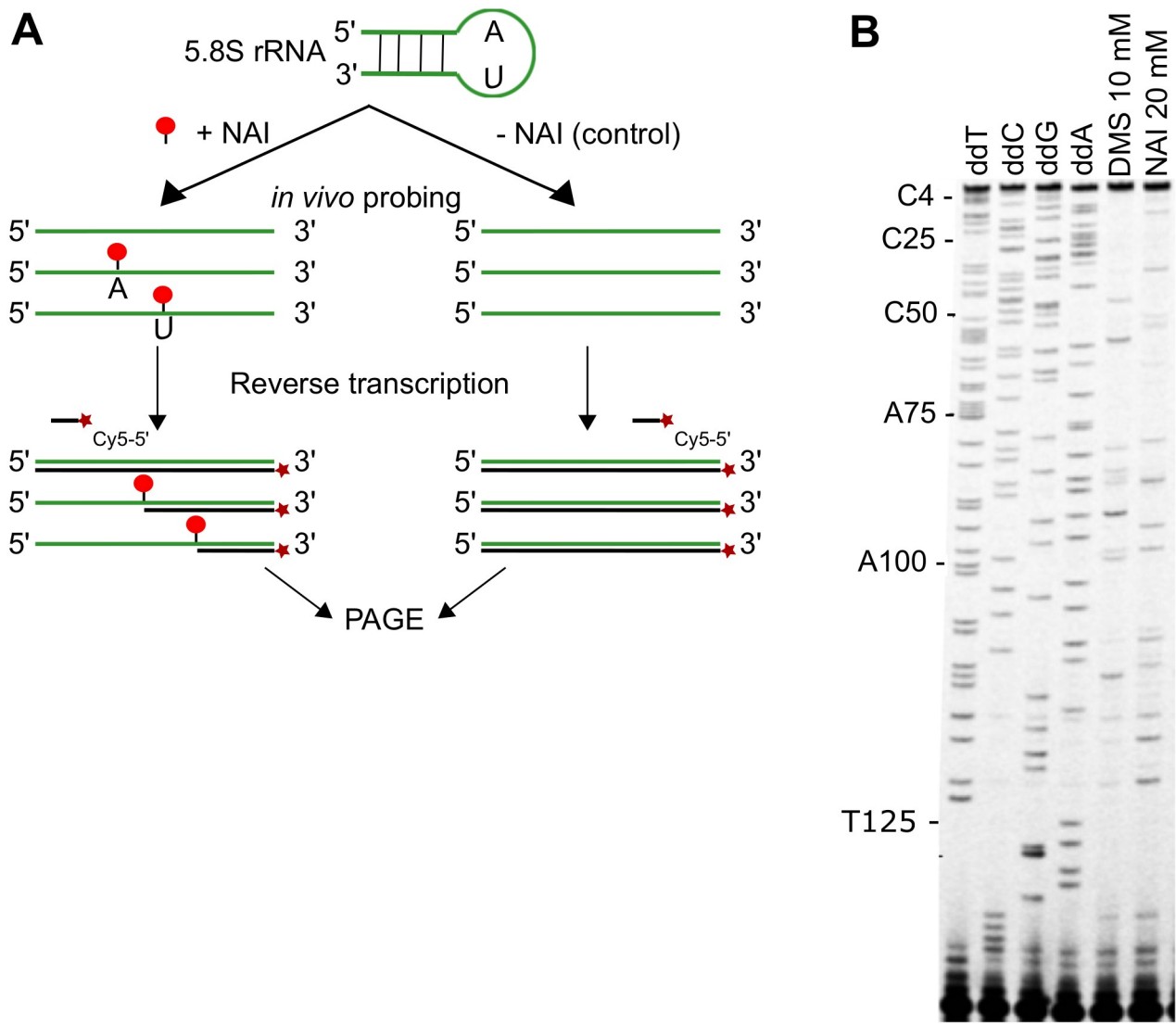

**Fig 1.** *In vivo* **probing of RNA structures in** *P. falciparum.* **(A)** Schematic of RNA structure probing. An asynchronous parasite culture was split into two, one exposed to NAI (+NAI) and the other exposed to DMSO as control (-NAI). Reverse transcription (RT) was performed using a 5'-Cy5 conjugated primer (dark red star) specific for the 5.8S rRNA of *P. falciparum* (in green). During RT, nucleotides that have been modified by NAI in the +NAI samples (red dots) will stall the reverse transcription, interrupting the elongation process (black line). Finally, elongation cDNA products are resolved on a PAGE gel. **(B)** Sequencing gel showing the 5.8S rRNA gene of *P. falciparum*. Lanes 1 to 4 are sequencing lanes where one of the four nucleotides in the reaction mix has been switched with the dideoxynucleotide, provoking RT-termination. Lanes 5 and 6 represent respectively DMS and NAI modifications on 5.8S rRNA.

were processed using *StructureFold2* and *RNAStructure* for experimental reactivity constrained folding [57] to determine the secondary structure of the RNA molecules. Each replicate for each chemical treatment was analysed separately. After preliminary analysis, because the distribution of structure differences between replicates was marginal and the correlation between replicates and pooled samples was good (S1 Fig), reads from both replicates were merged for subsequent analysis. To ensure structure calls of a high quality, mappings were filtered such that there were no more than 3 mismatches/indels from the reference and the first 5' base was matching. Coverage filtering also required transcripts with complete coverage between samples (i.e. meeting the *StructureFold2* default filtering threshold of an average

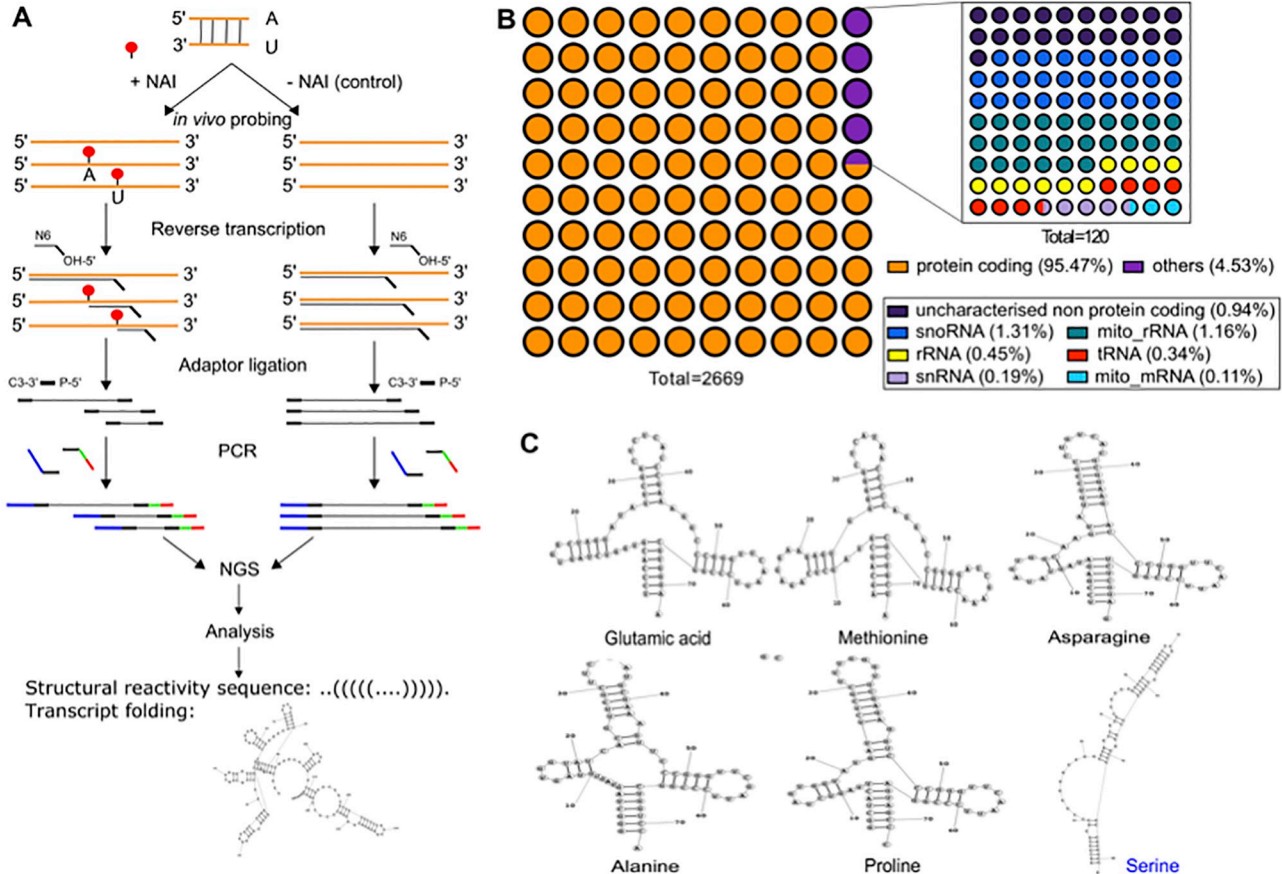

**Fig 2. RNA structurome of *P. falciparum* using Structure-seq. (A)** Schematic showing the Structure-seq pipeline. An asynchronous parasite culture was split into two, one exposed to NAI (+NAI) and the other exposed to DMSO as control (-NAI). Polyadenylated-enriched RNA fractions were then used to prepare the sequencing libraries in parallel. Fragmentation was performed to generate an average length (~250 nt) of RNA for sequencing followed by 3' dephosphorylation, which replaces the 2'3' cyclic phosphate group with a 3'-OH group for 3' adapter ligation. Then a 3' adapter (black line) was ligated to the 3'-OH group of dephosphorylated RNA. Next, RT of RNA molecules (orange line) was carried out using a designed reverse primer (dark blue line) at the 5' end. During RT, nucleotides that had been modified by NAI in the +NAI samples (red dots) would stall the reverse transcription, interrupting the elongation process (pink line). Later, a 5' adaptor (light blue line) was added by ssDNA ligation at the 3' end of the newly synthesised cDNA strand (pink line). Finally, forward primers (brown+light blue line) and reverse primers (dark blue+red+green line) with different indexes were added by PCR for NGS and bioinformatic analysis. Reads were aligned to the PF3D7 reference genome and subsequently analysed with *StructureFold2* to determine the structural reactivity sequence (transcribed as a dot/bracket sequence denoting paired/unpaired nucleotides), and with *RNAStructure* to predict the folding. **(B)** Representation of the diversity of the dataset regarding RNA families. **(C)** Structure-seq derived structures of glutamic acid (PF3D7_0411600), methionine (PF3D7_1339100), asparagine (PF3D7_0714700), alanine (PF3D7_0620800) and proline (PF3D7_1339200) tRNAs, all of which are similar to canonical tRNA structures, and of serine (PF3D7_041000) tRNA, which is non-canonical.

coverage >1.0 per nucleotide throughout the transcript (S2 Fig). Structural information was more comprehensive (and agreement between replicates was also better), for the NAI-probed datasets because all four unpaired nucleotides were detected. Subsequent analysis was therefore focused on the NAI dataset as the primary *in vivo* structurome.

## Structure of the NAI-probed dataset

After filtering for low-coverage genes, the NAI structurome covered 51.23% of the total transcriptome of *P. falciparum* (2699 transcripts detected out of 5268 total). We first looked at the representation of the various families of RNA molecules. Within our dataset 95.5% of the transcripts were protein coding transcripts, 1.27% corresponded to mitochondrial transcripts (1.16% mitochondrial rRNA, 0.11% mitochondrial mRNA) and 0.94% were uncharacterised

non-protein coding transcripts. Other types of non-coding RNA were also detected, 1.31% of snoRNA, 0.19% of snRNA and 0.34% of tRNA (Fig 2B). Within the protein coding transcripts, nine were encoded in the apicoplast, a subcellular organelle that is unique to apicomplexans and represents a relic chloroplast (S1 Table).

As well as the different families of transcripts, we detected 114 transcripts with splice variants (S2 Table). However, since we used a short-read technology to sequence the library, it was impossible to associate all the reads within a variant transcript to one variant. A second limitation of short-read sequencing was that it resulted in very limited coverage of untranslated regions, which in *P. falciparum* tend to be extremely A/U-biased (often over 90%), and also highly repetitive with many low-complexity regions. This precluded any comprehensive assessment of RNA structure in the UTRs versus CDSs of mRNAs. Nevertheless, this overall analysis indicates that our *in vivo* chemical probing, RNA extraction and polyA selection were robust because all the different RNA families in *P. falciparum* were detected, with a strong bias towards mRNAs, and if a long-read sequencing technique had been used, even the detection of transcript variants would likely have been possible.

Some RNA families have their function dictated by their structure. This is especially the case of tRNA. Within our dataset 9 tRNAs were identified. Using *RNAStructure* we were able to call their structures and 5 were similar to canonical eukaryotic tRNA molecules. Those tRNAs were for glutamic acid (PF3D7_0411600, PF3D7_0527700), methionine (PF3D7_1339100), asparagine (PF3D7_0714700), alanine (PF3D7_0620800) and proline (PF3D7_1339200) (Fig 2C). In the other 3 tRNAs, for valine (PF3D7_0312600), serine (PF3D7_0410100) and leucine (PF3D7_0620900), the lowest-free-energy structures predicted by *RNAStructure* were not canonical (S3 Fig). For some but not all of these, alternative folding with less favourable energy produced canonical structures, albeit less compatible with the data defining free/flexible bases (e.g. S3A Fig). It is theoretically possible that dynamic structural variants of some tRNAs exist *in vivo*, although it is unclear why some tRNAs would differ from others in the dataset. Importantly, however, tRNA structure is also dependent on nucleotide base modification such as $m^2A$, $m^2G$, pseudouridine, etc. [46]. Those modifications have not been assessed in this study, but it is possible that a) they would influence the apparently-noncanonical tRNA structures and b) they might cause extra RT-stalling and thus appear erroneously as 'probed' bases, impeding the determination of the true structures of hyper-modified RNAs such as tRNAs.

In general, opportunities to 'verify' the structurome were limited because of the lack of *Plasmodium* RNA structures obtained independently, by orthogonal experimental methods. The canonical structural model of a blood-stage *P. falciparum* rRNA is available from the Comparative RNA Web (CRW) site and this was used to map base reactivities obtained *in vivo* (S4 Fig). The result demonstrates visually the A/C reactivity of DMS versus the all-base reactivity of NAI, and makes it clear that many bases modelled as unpaired are nevertheless not highly reactive *in vivo*. Other factors such as protein binding and base modification probably affect their reactivity, emphasising the over-simplification imposed by *in silico* modelling and the extra level of information obtained by *in vivo* structural probing.

## Divergence between *in vivo* RNA structures and structures modelled *in silico*

Using the *RNAStructure* algorithm, RNA structures can be predicted from primary sequence alone, or they can be predicted with the addition of base-pairing information gained experimentally. We generated both types of structure, transcriptome-wide, taking only the lowest-free-energy structure in all cases where the algorithm generated several alternatives. This was

**Table 2. Distribution of 'divergence' normalised by transcript length between base pairing determined by Structure-seq and *in silico*.**

|  | Protein coding | mito_mRNA | Uncharacterized non protein coding | snoRNA | tRNA | rRNA | snRNA | mito_ rRNA |
|---|---|---|---|---|---|---|---|---|
| # transcript | 2549 | 3 | 25 | 35 | 9 | 8 | 5 | 36 |
| Median | 0.71 | 0.81 | 0.53 | 0.44 | 0.17 | 0.22 | 0.46 | 0.54 |
| Minimum | 0.04 | 0.75 | 0.07 | 0.08 | 0.06 | 0.02 | 0.19 | 0.00 |
| 25% Percentile | 0.60 | 0.75 | 0.34 | 0.26 | 0.10 | 0.06 | 0.19 | 0.35 |
| 75% Percentile | 0.79 | 0.89 | 0.69 | 0.63 | 0.60 | 0.35 | 0.56 | 0.74 |
| Maximum | 1.03 | 0.89 | 0.91 | 0.92 | 0.73 | 0.47 | 0.66 | 1.05 |

Data are categorised by family of RNA.

simply because, in the absence of other constraints, the lowest-free-energy structure is the most likely to form.

Firstly, we sought to discover how much these two types of structures would differ in *P. falciparum*, i.e. how much new information was gained by performing the *in vivo* structurome? We calculated the ratio of bases that differed (i.e. that were paired versus unpaired) between each *in-silico*-predicted and probing-informed structure, and termed this ratio the 'divergence'. Most of the non-protein coding transcripts had relatively low divergence, meaning that they were minimally informed by probing (Table 2). This was expected for highly conserved, stereotypical structures like rRNAs and tRNAs—although the divergence for mitochondrial rRNAs, which are encoded in a highly fragmentary fashion in the *P. falciparum* mitochondrial genome, was notably greater than for nuclear-encoded rRNAs. By contrast, the divergence of protein coding transcripts was much higher: a median of 0.7067 for the nuclear-encoded mRNAs and 0.8117 for mitochondrial mRNAs (Table 2).

Since structure and function in non-protein coding transcripts, like tRNAs, are deeply intertwined, we then sought any relationship between folding complexity and biological function. Using the 'divergence' as a metric, we split the protein coding transcripts into 3 tiers: minimal value to 25th percentile, 25th to 75th percentile and 75th percentile to the highest value (Table 2). The list of transcripts was then submitted for gene ontology enrichment and using Revigo [58] we obtained broad families of GO terms and networks of terms belonging to a similar biological pathway. In tier 1, with the lowest 'divergence', we observed an enrichment of 21 terms, with the strongest enrichment in terms related to ribosomal functions (see GO term 7, 'translation', in Fig 3A, S3 Table). This was expected, since ribosomal RNA structures are highly conserved and are likely to be predicted accurately *in silico* with little alteration in *P. falciparum*. Twenty-one terms were found in tier 2. Two terms were again strongly enriched: 'translation' and 'cellular amide metabolism', and these fell into a loosely-associated collection of similar terms around 'protein' and 'protein regulation'. There was a second cluster restricted to two terms around RNA splicing, and a third cluster around mitochondrian/organellar organisation (Fig 3B, S3 Table). In tier 3, with the lowest level of identity between *in vivo* structures and *in silico* predictions, only seven terms were enriched and there was only one cluster, composed of four terms strongly enriched around different kinds of amino acid metabolism (Fig 3C, S3 Table).

Secondly, we assessed the value of performing Structure-seq instead of just *in silico* prediction for modelling the shape of RNA molecules. We used ViennaFold on the *P. falciparum* transcriptome to fold the structures with or without reactivity data. We first aligned the dot/bracket sequences obtained from *in silico* folding with the Structure-seq-determined sequence of the U2 snRNA (PF3D7_1137000). It showed a difference in the pairing of 74 nucleotides out of 198, meaning a 37.4% discrepancy in the structure determined *in silico* compared to *in*

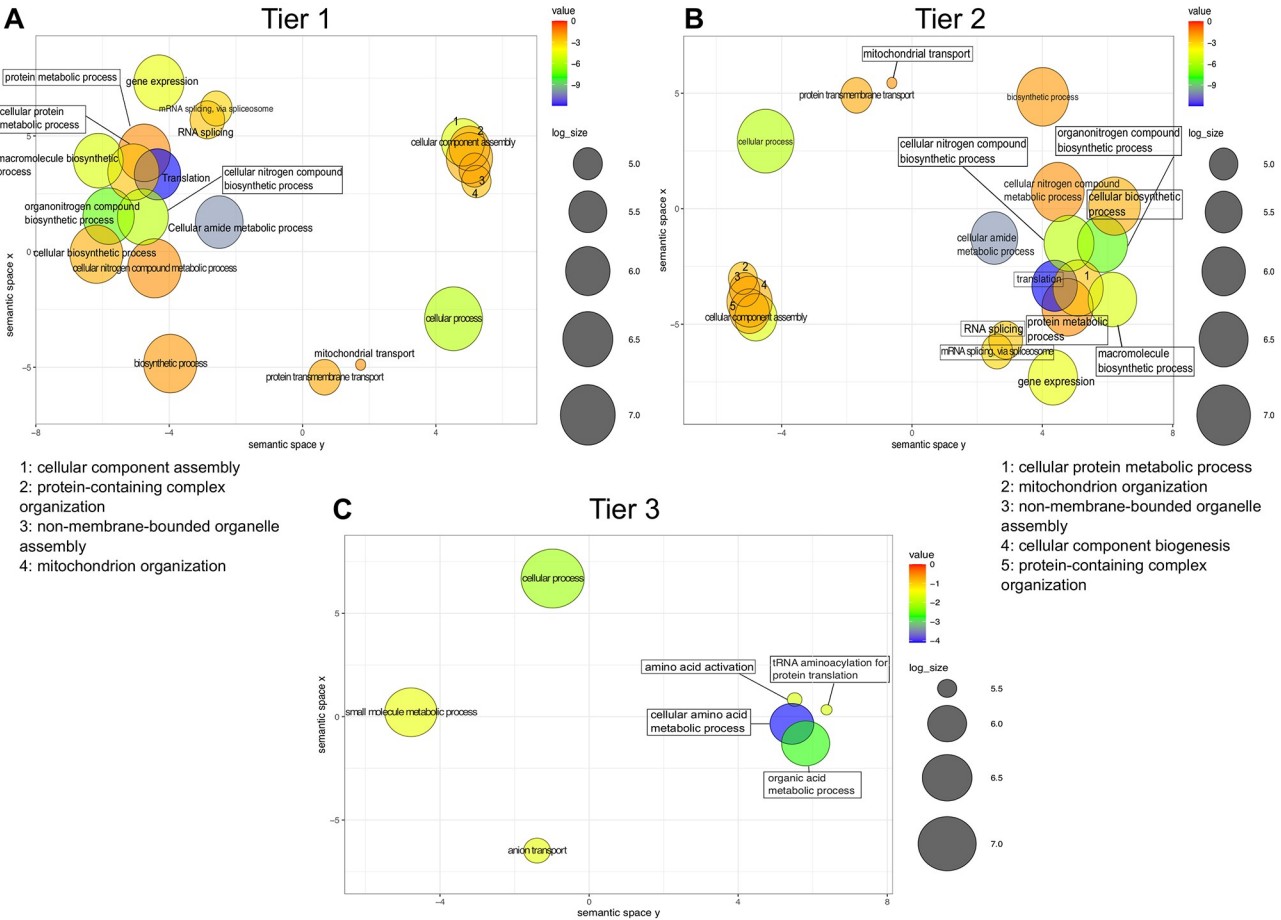

**Fig 3. GO terms enrichment scatterplot for protein coding transcripts, tier 1 to 3.** All plots represent the GO enrichment provided by REVIGO for (**A**) tier 1, (**B**) tier 2 and (**C**) tier 3. Detailed values are available in S3 Table. Each GO term is represented by a numbered circle on the plot, with placement indicating its similarity to nearby GO terms. The colour scale represents the adjusted p-value of the GO term and the size of each circle represents the log10 of the number of annotations per GO term.

*vivo* (Fig 4A). Structurally, the difference between the two transcripts is major and only 3 loops are common to both structures (Fig 4B). More interestingly, an RNA-appropriate measurement of Minimum Free Energy, MFEden [59], revealed that the Structure-seq U2 snRNA structure had a lower MFEden (-24.76 versus 15.50). Extending this to the whole dataset, we observed a similar effect in each RNA category (Table 3). Thus, structures called from Structure-seq are more likely to form and are more stable than the ones predicted only using an *in silico* prediction.

## Effect of A/U bias in the structurome of *P. falciparum* by comparison with an A/U-balanced *Plasmodium* species

The *Plasmodium* genus offers the possibility to compare very different genome compositions. The genome of *P. falciparum* is ~80% A/T rich while *P. knowlesi*, a macaque *Plasmodium* species which also infects humans, is 62.5% A/T rich [60]. In order to assess whether the base composition of the genome, and hence the transcriptome, could affect the structurome, we compared the *in silico* folding of *P. falciparum* and *P. knowlesi* structuromes. Firstly, we assessed the stability of the two predicted structuromes by comparing the average free energy

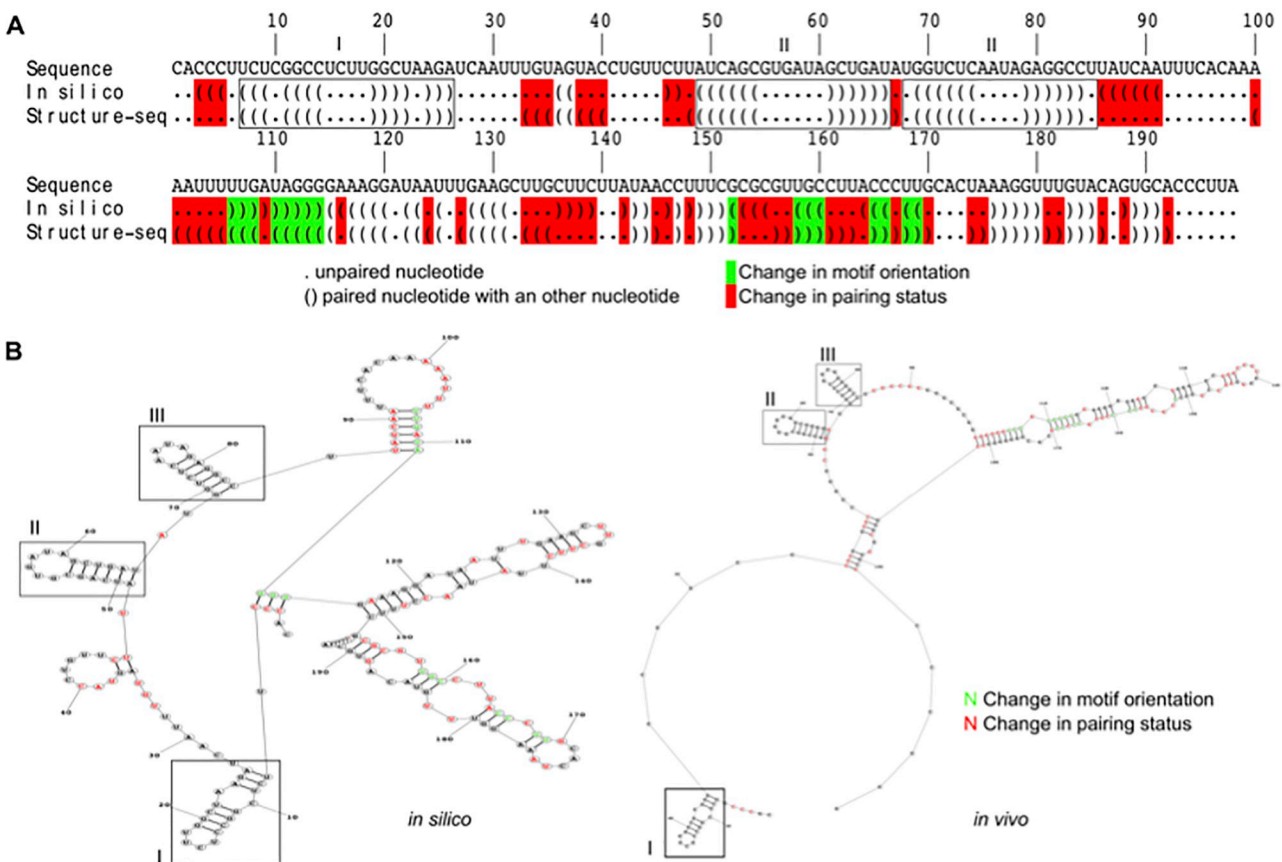

**Fig 4. Difference in U2 snRNA structure determined by Structure-seq versus *in silico* prediction. (A)** U2 snRNA (PF3D7_1137000) reactivity sequence alignment, comparing *in vitro* probing using NAI and *in silico* prediction using the same constraint parameters. The first line shows the nucleotide sequence, the second line shows the *in silico* predicted dot/bracket sequence and the third line, the Structure-seq dot/bracket sequence extracted from reactivity measurement. A dot shows a nucleotide not paired with any other nucleotide. An open bracket shows a nucleotide paired with another and starting a structural motif, while a closed bracket shows a nucleotide paired with another but finishing a structural motif. Red-highlighted dot or bracket indicates a change of pairing between the NAI and *in silico* predicted structural sequence. A green-highlighted bracket indicates a change in motif orientation between the NAI and *in silico* predicted structural sequence. Boxes with roman numerals indicate the conserved structures between the two shapes. **(B)** U2 snRNA structure predicted *in silico* from RNA sequence alone, on the left, and with the addition of reactivity data on the right. The red and green colouring have the same meaning as in (A).

(MFEden [59]) of all folded structures: *P. falciparum* had a significantly higher MFEden than *P. knowlesi* (respectively 14.03 and 7.1, t-test *p*-value <0.0001). Secondly, we looked at the occurrence of various structures by comparing the length-normalised number of nucleotides participating in a structure (e.g. a hairpin or bulge) in each *P. falciparum* transcript with their *P. knowlesi* orthologs. The structurome of *P. falciparum* had significantly more nucleotides involved in hairpins and multi-stem loops than the structurome of *P. knowlesi* (t-test, *p*-value <0.0001), whereas *P. knowlesi* had significantly more nucleotides involved in stems and bulges (t-test, *p*-value <0.0001) (S5 Table).

**Table 3. MFEden average for every type of RNA molecule.**

|  | protein coding | mito_mRNA | uncharacterised non protein coding | snoRNA | tRNA | rRNA | snRNA | mito_rRNA |
|---|---|---|---|---|---|---|---|---|
| Structure-seq | -20.5 | -27.0 | -29.8 | -29.6 | -63 | -31.1 | -37.9 | -25 |
| *in silico* | 14.1 | 10.3 | 10.7 | 10.2 | -8.8 | 3.2 | 10.7 | 2.7 |

All MFEden values per transcript are in S4 Table.

## RNA structures affect translation efficiency in a stage specific manner

Fig 2 shows that various families of RNAs display very conserved structures that are related to their particular functions. This is clearly true for structural RNAs (tRNAs, rRNAs, etc.), but it may also be true for mRNAs, whose main function is to be translated into proteins. The stability of the transcript and the efficiency of its translation could both be affected by mRNA structure. Therefore, if mRNA structures can vary with varying cellular conditions, transcript and protein abundance may both vary accordingly.

We first explored the concept that transcript abundance could vary with cellular conditions due to changes in mRNA structure. For this, we used published microarray and transcriptomic datasets from *P. falciparum* under physiological stresses like hyperoxia [51], or drug exposure like chloroquine treatment [52]. These datasets reveal which transcripts are upregulated and downregulated in each condition. We compared the most differentially expressed genes in each condition with the structurome dataset, to see if they were particularly structured or unstructured. Neither the overall degree of 'divergence' nor the representation of diverse structures (the proportion of hairpins, loops, etc), correlated with differential expression under stress (S6 Table).

Secondly, we looked at the effect of RNA structures on translation efficiency (TE). To establish the translation efficiency for each transcript in our dataset, we extracted from previously published work [53] the average level of ribosome attachment per transcript, determined by Ribo-seq. We obtained TE for the transcriptome of rings, early and late trophozoites and schizonts. Applying a linear regression model, we compared TE with transcript structuredness ('NAI ratio', i.e. the ratio of paired to unpaired bases) and also the proportion of different motifs such as stems, hairpins, multi-loop stems and bulges (Fig 5). We observed no correlation between the transcript-length-normalised 'structuredness' and TE in any of the four stages (Fig 5A). Therefore, there was no overall relationship between the proportion of paired bases in a transcript and the efficiency of its translation. However, breaking down the different possible structures (Fig 5B–5E) we observed a positive correlation of TE with the presence of stems and a negative correlation between TE and hairpins, multi-loop stems and bulges. The correlations were only seen in the more mature parasite stages, late trophozoites and schizonts.

## RNA decay is mediated by the same structures across the cell cycle

The final critical part of an RNA molecule's life inside the cell is its rate of decay. Following a similar reasoning as for translation efficiency, we extracted RNA decay values from data published by Painter *et al.* [54] and tested for any correlation between RNA decay and RNA secondary structures in the different cell cycle stages (rings, early and late trophozoites and schizonts). For this, we applied a linear regression model to compare RNA decay (the higher the value, the less stable the molecule is) to the number of different structures per transcript. We observed no correlation between RNA decay and RNA secondary structures involving high amounts of base pairing: stems and hairpins (Fig 6A and 6B). However, when nucleotides were more exposed, as in bulges and multi-stem loops (Fig 6C and 6D) we observed correlations between RNA decay and the presence of those structures in all stages (except for multi-stem loops in ring-stage) (Fig 6C).

## Discussion

In this study, we produced the SHAPE-seq-determined structurome of a parasitic protozoan, *P. falciparum*. This had never previously been performed for any protozoan parasite, although a second study, appearing after the submission of this manuscript, has also now described an *in vivo P. falciparum* structurome [61]. This study covered only 23–35% of the transcriptome

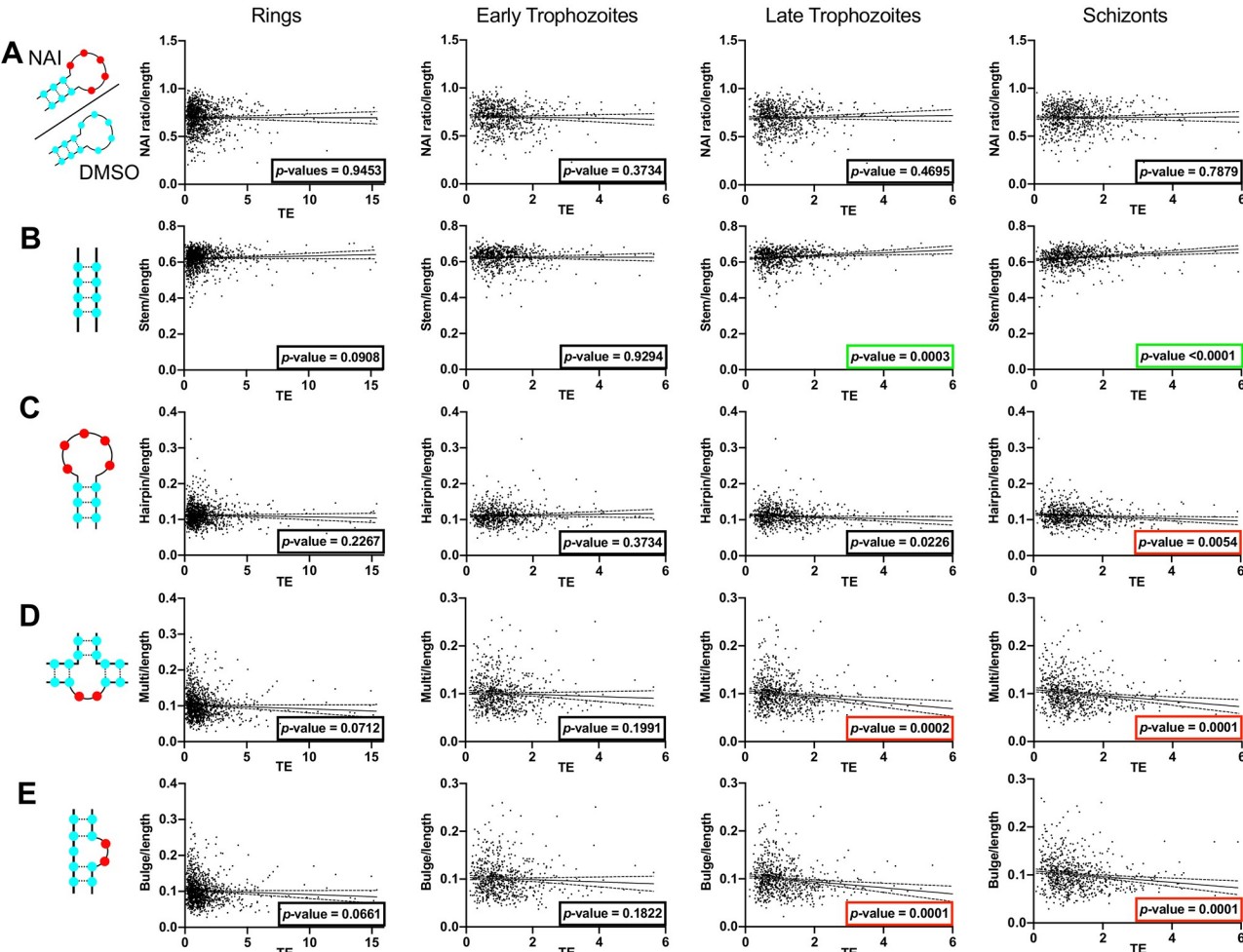

**Fig 5. Impact of various RNA motifs on transcription efficiency across the erythrocytic cycle of *P. falciparum*.** The columns represent the different erythrocytic stages of *P. falciparum* and the rows represent the correlation between transcription efficiency and **(A)** the transcript structuredness normalised by the transcript length, **(B)** the number of stems, **(C)** the number of hairpins, **(D)** the number of multi-stem loops and **(E)** the number of bulges. Statistically significant positive correlations have a *p*-value framed in green and statistically significant negative correlations have a *p*-value framed in red.

(in trophozoites and rings respectively), compared to >51% here, and had approximately 35% fewer reads per sample with markedly lower mappability of reads. Our study therefore remains the most comprehensive structurome in *P. falciparum* to date. A third study, equally recently published, mapped duplex RNA regions by RNA-seq after degradation of ssRNA [62]; however this does not produce data directly comparable to an NAI-probed *in vivo* structurome.

Firstly, we demonstrated that it is possible to use probing chemicals on the parasite within the host erythrocyte, meaning that the chemicals can get through the host cell membrane, the parasitophorous vacuole and the parasite membrane. The fact that we could also detect mitochondrial and apicoplast transcripts shows that the chemicals also passed through the inner organellar membranes. Thus, we obtained *in vivo* structures for about half the transcripts in the transcriptome, including structural and noncoding RNAs as well as mRNAs. Most of the structural RNAs had stereotypical structures, although a subset of tRNAs did not model stereotypically—an interesting observation that merits further investigation. Amongst mRNAs, the *in vivo* structures tended to have lower free energy, making them more likely to form, than the

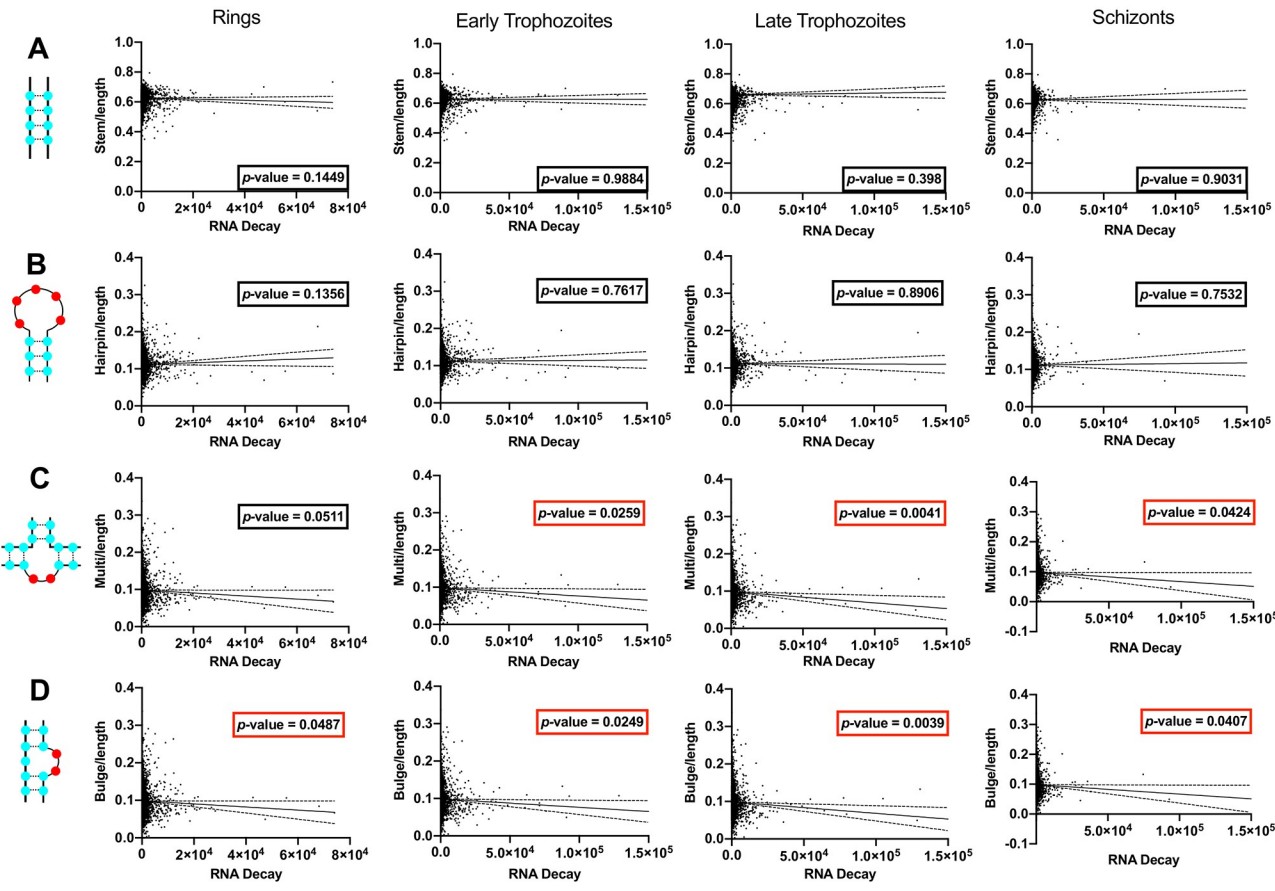

**Fig 6. Correlation between RNA secondary structures and RNA decay across the erythocytic cycle of *P. falciparum*.** The columns represent the different erythrocytic stages of *P. falciparum* and the rows represent the correlation between RNA decay and **(A)** the number of stems, **(B)** the number of hairpins, **(C)** the number of multi-stem loops and **(D)** the number of bulges. Statistically significant negative correlations have a *p*-value framed in red.

equivalent structures predicted *in silico*, thus demonstrating the value of the experimental approach. Highly conserved RNAs such as ribosomal RNAs tended to be predicted well, whereas classes of genes whose structures were poorly predicted *in silico* included genes involved in metabolism, biosynthesis of complex molecules, and transport. These genes may have become particularly divergent in *P. falciparum* due to its parasitic intracellular lifestyle.

We also took advantage of the striking disparities in genome composition in the *Plasmodium* genus to see if the extreme base composition of the *P. falciparum* transcriptome might have an impact on the different structures formed, by comparing the predicted structurome of *P. knowlesi* with that of *P. falciparum*. We observed that in *P. falciparum* transcripts were more prone to form structures like hairpins and multi-stem loops, whereas a more balanced genome like *P. knowlesi* gave rise to orthologous transcripts with more stems. Overall, *P. falciparum* transcripts were predicted to fold with higher free energy than their orthologs in *P. knowlesi*, suggesting that an A/U-biased transcriptome is less stable. However, this might be mitigated by *in vivo* factors, since we also found that overall the *P. falciparum in vivo* structurome had lower free energy than was predicted *in silico*. It would be interesting in future to perform an *in vivo* structurome of *P. knowlesi* and see whether *in vivo* transcripts tend towards greater stability in this species as well.

To examine how the RNA structurome might influence parasite biology, we looked at the impacts of the different structures on translation efficiency and RNA decay. We showed that RNA structures had no impact on translation efficiency in rings and early trophozoites, two stages of the erythrocytic cycle that are translationally less active [53, 63]. On the contrary, late trophozoites and schizonts are more translationally active and negative correlations between translation efficiency and complex structures like hairpins, multi-loop stem and bulges were observed. Recently, we studied a particularly stable RNA structure in *P. falciparum*, the RNA G-quadruplex, and demonstrated that it likewise had a negative impact on translation [64]. We also observed a positive correlation between translation efficiency and a simpler RNA structure, stems. This is consistent with the idea that stem structures can facilitate ribosome attachment, whereas other complex structures can inhibit it [65].

The correlations described above were not detected in either of the parallel *P. falciparum* 'structurome' studies. A study comparing duplex RNA regions with translation efficiency failed to observe any clear correlations [62], probably because it did not distinguish the subtleties of different types of RNA secondary structure: this highlights the power of the structurome approach taken here. Meanwhile, the comparable study by Wang and co-workers [61] reported that the least-structured transcripts had the highest abundance, but since abundance was derived only from a snapshot RNA-seq measurement, it was not possible to attribute this specifically to transcript half-life, decay rate, etc. as individual factors that might be affected by RNA structure. Our study examined two such factors individually—translation efficiency and also RNA decay rate—and therefore generated different insights.

There was a negative correlation between structures and RNA decay, so we can hypothesise that RNA structures influence protein expression on at least two levels: the rate of translation and the degree of transcript stability. Similar effects were previously demonstrated in human cell lines, and furthermore RNA base modifications were also found to be important, impacting protein expression by changing the functional half-life of transcripts [66]. This latter area has been little explored in *P. falciparum* as yet, but a recent study of base modification in *P. falciparum* likewise implicated RNA base modification in protein expression [67].

This study focuses on RNA structure in the erythrocytic life cycle of *Plasmodium* parasites. Two more parts of the full life cycle remain to be studied in future: the liver stage and the sexual cycle which happens in the host vector, a mosquito. The liver stages are particularly interesting because two human-infective *Plasmodium* species, *P. vivax* and *P. ovale*, can make hypnozoites—a dormant form that can stay inside the liver of infected people for months and then reactivate to cause disease [68]. Currently the biology of hypnozoites remains complicated to investigate. However, translational repression may be key to long-term dormancy, so the RNA structurome of hypnozoites could be intriguing. A more accessible stage, the gametocyte, also shows dormancy (albeit a dormancy that lasts days rather than months), and female gametocytes are known to translationally repress hundreds of transcripts [69]. Once in the mosquito midgut, the translational repression is lifted, and we can hypothesise that the change of temperature between a warm-blooded host and a non-thermoregulatory insect would impact RNA structures, playing a role in lifting the repression.

Within the mosquito, the sexual life cycle encompasses four different life stages for which no RNA structures are known. This step of the life cycle is crucially important in completion of the transmission chain and also in generating genetic diversity [70]. Recent progress in single cell RNA sequencing give us the opportunity to look at the transcriptome of single parasites at all life cycle stages, including those in the host vector [71]. In parallel, novel RNA structure determination techniques, like SHAPE-MaP-seq associated with the latest algorithms like DRACO, would make the determination of single cell RNA structures possible [72]. Other technological breakthroughs like SMRT-seq (Single Molecule, Real Time) associated with

SHAPE-MaP would allow access to transcriptional variant structures [73], which were not possible to analyse in this study, and this could shed new light on post-transcriptional regulation in *Plasmodium*. Finally, the key roles of RNA-binding proteins in determining RNA structures *in vivo* have yet to be examined. Broadly, Wang and co-workers found that de-proteinised RNAs that had been refolded *in vitro* were less structured, and/or protein-bound, than the same RNAs *in vivo*, which is not unexpected [61]. However, more specific future experiments could potentially knock out single RNA binding proteins and then compare the resultant RNA structures *in vivo*. This would shed more detailed light on the determinants of the *P. falciparum* RNA structurome.

Overall, this work paves the way to decipher another level of complexity in the molecular biology of *Plasmodium* parasites.

## Supporting information

**S1 Raw images. Polyacrylamide gels showing the reverse-transcribed 5.8S rRNA gene of *P. falciparum*.** (A) results after probing with DMS (2 concentrations), NAI, vehicle control, i.e. DMSO, or no treatment (UT). DMS and NAI modifications are detected as reverse-transcriptase stops, with very similar patterns at both DMS concentrations. The vehicle control confirms that there is negligible 'background', i.e. almost no reverse-transcriptase stops, yielding a near-empty lane similar to the untreated control. (B) full-width gel that appears cropped (as highlighted in red) in Fig 1B, visualised with Fujifilm FLA 9000.
(PDF)

**S1 Fig. DMS and NAI samples: Correlation and pooling data.** (A) For each of the different treatments (both individual replicates and pooled), the distribution of 'RNAdistance' is plotted, i.e. the computed distances between the purely computationally-determined structures and those guided by reactivity data. The y-axis represents the frequency of RNAdistance metrics, which are represented on the x-axis. Replicates of each treatment appear similar, as quantified in (B). (RNAdistance calculated as per: R. Lorenz, S.H. Bernhart, C. Hoener zu Siederdissen, H. Tafer, C. Flamm, P.F. Stadler and I.L. Hofacker (2011), "ViennaRNA Package 2.0", Algorithms for Molecular Biology: 6:26). B) Pearson correlation heat map showing how the RNAdistance-derived distributions in panel A correlate across samples. Two distinct groupings are clearly formed for NAI and DMS.
(TIFF)

**S2 Fig. Plots showing the coverage of each transcript (averaged per nucleotide) in each replicate of each treatment condition.** More than 50% of transcripts in every condition met the coverage threshold of >1.0 per nucleotide.
(TIF)

**S3 Fig. *RNAStructure* folding of three non-canonical-shaped tRNAs.** (A) valine tRNA (PF3D7_0312600), (B) serine tRNA (PF3D7_0410100) and (C) leucine (PF3D7_0620900). The structures in (A) and (C) could also be folded into a less-energetically favourable canonical shape: to exemplify this, NAI-reactive bases in (A) are marked on both structures (red, orange, green from highest to lowest relative reactivity). The structure in B could not be folded into a canonical shape, even at 50% less favourable energy, when using the constraints obtained by NAI probing.
(TIFF)

**S4 Fig. *P. falciparum* 18S rRNA structure.** The canonical structure of the *P. falciparum* 18S rRNA (from https://crw-site.chemistry.gatech.edu/) was compared with base reactivity data

for the assembled sequence of the PF3D7_0725600 gene (encoding blood-stage-expressed 18S rRNA). Maximum base reactivities for both the NAI and DMS datasets were mapped: NAI reactivities in the blue channel and DMS reactivities in the red channel, hence dually-reactive bases appear pink. Reactivities were scaled to colour intensity.
(TIFF)

**S1 Table. Complete and detailed list of transcripts found during the Structure-seq analysis.** The file contains all the genomic information, product description, gene type, transcript length, GC and AT content, reactivity for each replicated of DMS and NAI experiment plus the combined one, and the gene ontology information.
(XLSX)

**S2 Table. Table of transcript variants.** Same information available as in S1 Table.
(XLSX)

**S3 Table. REVIGO output for clustering of GO terms.** Each tab represents the data per tier of divergence.
(XLSX)

**S4 Table. Table of free energy and transcript structures from Structure-seq and *in silico* analysis.**
(XLSX)

**S5 Table. List of the *P. falciparum* transcripts and their *P. knowlesi* orthologs with *in silico* folding information normalised by transcript length.**
(XLSX)

**S6 Table. Table of raw data from 2 previous studies on perturbation of the transcriptome via environmental conditions.** Sheet 1 and 2, respectively named 4h_hyperoxia and 8h_hyperoxia, are the raw microarray data from Torrentino-Madamet *et al*. [51]. Sheet 3, named CQ_dataset, is the list of differentially expressed genes after long chloroquine treatment from Untaroiu *et al*. [52]. Original data have a column header in red, all column headers in black were added for the purpose of this study. Rows 1 and 2 show the average value for each category for up- and down- regulated genes.
(XLSX)

## Acknowledgments

We would like to thank Dr. Betty Chung from the Cambridge Pathology Department for her help with the transcription efficiency data processing.

## Author Contributions

**Conceptualization:** Chun Kit Kwok, Catherine J. Merrick.

**Data curation:** Franck Dumetz, Anton J. Enright.

**Formal analysis:** Franck Dumetz, Anton J. Enright.

**Funding acquisition:** Chun Kit Kwok, Catherine J. Merrick.

**Investigation:** Franck Dumetz, Jieyu Zhao.

**Methodology:** Franck Dumetz, Chun Kit Kwok.

**Project administration:** Catherine J. Merrick.

**Supervision:** Chun Kit Kwok, Catherine J. Merrick.

**Visualization:** Franck Dumetz, Anton J. Enright.

**Writing – original draft:** Franck Dumetz, Jieyu Zhao.

**Writing – review & editing:** Chun Kit Kwok, Catherine J. Merrick.

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
