## [Decision Letter · Decision Letter 0]

29 Apr 2022

PONE-D-22-07594The in vivo RNA structurome of the malaria parasite Plasmodium falciparum, a protozoan with an A/T-rich transcriptomePLOS ONE

Dear Dr. Merrick,

Thank you for submitting your manuscript to PLOS ONE. After careful consideration, we feel that it has merit but does not fully meet PLOS ONE’s publication criteria as it currently stands. Therefore, we invite you to submit a revised version of the manuscript that addresses the points raised during the review process.

We look forward to receiving your revised manuscript.

Kind regards,

Friedrich Frischknecht

Academic Editor

PLOS ONE

Journal Requirements:

"UK Medical Research Council (grants MR/K000535/1 and MR/L008823/1) to CJM, Royal Society Kan Tong Po fellowship, Shenzhen Basic Research Project (JCYJ20180507181642811), Research Grants Council of the Hong Kong SAR, China Projects CityU 11101519, CityU 11100218, N_CityU110/17, CityU 21302317, Croucher Foundation Project No. 9509003, 9500030, State Key Laboratory of Marine Pollution Director Discretionary Fund, City University of Hong Kong projects 6000711, 7005503, 9680261, 9667222 to CKK. The funders had no role in study design, data collection and interpretation, or the decision to submit the work for publication."

We note that you have provided funding information that is not currently declared in your Funding Statement. However, funding information should not appear in the Funding section or other areas of your manuscript. We will only publish funding information present in the Funding Statement section of the online submission form. 

Additional Editor Comments:

Dear Catherine,

thanks for sending your nice paper to Plos One. It has now been seen by two experts in the field, who, as you will see, liked the work but have some suggestions to improve the paper. I would encourage you to revise the manuscript according to their comments.

Sincerely

Freddy

Reviewers' comments:

Reviewer's Responses to Questions

**Comments to the Author**

1. Is the manuscript technically sound, and do the data support the conclusions?

Reviewer #1: Yes

Reviewer #2: Yes

2. Has the statistical analysis been performed appropriately and rigorously? 

Reviewer #1: Yes

Reviewer #2: Yes

3. Have the authors made all data underlying the findings in their manuscript fully available?

Reviewer #1: No

Reviewer #2: Yes

4. Is the manuscript presented in an intelligible fashion and written in standard English?

Reviewer #1: Yes

Reviewer #2: Yes

5. Review Comments to the Author

Reviewer #1: In this manuscript Dumetz et al. probe RNA secondary structures during the asexual replication of Plasmodium falciparum within human red blood cells. They use the well-established method Strucutre-seq that uses the chemical NAI to probe unpaired nucleotides and subsequent RT fall-off to interrogate RNA secondary structures on a transcriptome wide scale. Comparisons of their in vivo derived structurs to known, well characterized structures (e.g. tRNA) show the robustness of their method and I particularly like the ‘divergence’ score to categorize different RNA classes. Finally, the authors compare their RNA structure to translational efficiencies and RNA decay rates and find that some structures within RNA transcripts can influence those. Overall, the study is well designed, the methodology is clear, their findings well-presented and the study will overall lay the foundation to study dynamics of the RNA structurome across the P. falciparum life-cycle in the future.

Please find my comments to specific points below:

line 344ff: The structure-seq relies on RT fall-off. This also happens very frequently at sites of RNA modifications, especially in hypermodified molecules such as tRNA. For the tRNA where the authors do not find a ‘canonical’ structure, the RT fall-off because of modifications might make the probing of the actual structure impossible. This information should be added to the Discussion.

What about different structures in different regions of mRNA transcripts, i.e. 5’UTR-CDS-3’UTR. UTRs in P. falciparum are substantially more A/U rich than the CDS. Please provide a paragraph/table/figure showing where different structures are located and if there is a potential bias.

line 361: The authors do mention the previously published P. falciparum structuromes. How similar are the structures especially of mRNAs identified in the different studies?

line 325: It is not fully clear if all presented structures are derived from the NAI samples. If so, why did the authors chose to present only those in the main text although all samples where sequenced and present in the supplement? How different are the structures of the NAI and DMS samples?

line 502: Different transcript regions have substantially different functions within a mRNA molecule. 5’UTRs are essential for translation and can even initiate protein synthesis cap-idenpendent with secondary structures, i.e. IRES. On the otherhand 3’UTR are crucial for RNA decay. So how does the comparison with translational efficiencies and mRNA decay change when focusing only on structures on certain regions, i.e. 5’UTR and translation and 3’UTR and decay?

Are there certain conserved structures in particular regions present in multiple different transcripts?

Please upload all raw and processed sequencing data to the SRA and GEO databases.

Minor

Please ensure to accurately state “A/U” when talking about the transcriptome (e.g. in the title)

Reviewer #2: Merrick and co-workers used a combination of chemical modification and RNA seq to inform on the secondary structure of mRNAs of the malaria-causing parasite Plasmodium falciparum. The authors were able to provide data on more than half of the parasite’s transcriptome and compare a AT-rich to a AT-balanced genome. The current study complements and expands insights from two recent studies on the same subject.

The presented data appear solid and the conclusions are scientifically sound. But some minor changes (for example Fig. 3) would help the reader to better access the significance of this work. Below are some specific minor comments, which should help to further strengthen this manuscript.

Line 214: Technically the parasite does not live “entirely” inside a host cell, many stages are extracellularly. Please revise.

Sentence starting in line 250 may be clearer if stating that nucleotides were unpaired before labelling reaction.

The labels of Suppl. Fig. 3 are very small.

For Fig. 2C, would it be possible to have the experimentally-determined structures superimposed on the “canonical” folding? This would strengthen the point. While Fig. 2C and Suppl. Fig. 4 clearly show the strength of the approach in detecting the expected canonical structures of some tRNAs, showing some of the non-canonical structures in the main figure would be appreciated.

Revise formatting of the reference in line 358.

End of the sentence in line 403: . instead of ,

Fig. 3, please provide a scale for the different shades of red/fold enrichment. Which P-value cut-off was used to identify significantly enriched GO terms? Also, the figure would be easier to understand, if panels A, B, and C had labels indicating the divergence tiers. It is unclear how the numbers of enriched GO terms in the text correspond to the number of GO terms shown in the figure, please explain.

The Fig. 4A would benefit from a legend describing what the green/red shading and the boxes indicate. Please also explain here what the dot/bracket sequences mean.

Is the term “significant” (line 435) used to indicate statistical significance? If so, please provide information on the tests. If not, please rephrase.

Fig. 5 and 6 caption: It is not the “life cycle” but the development inside erythrocytes.

While the other two studies investigating the structure of P. falciparum mRNAs are mentioned, this manuscript would benefit from an extended discussion of the results from the other studies.

6. PLOS authors have the option to publish the peer review history of their article (what does this mean?). If published, this will include your full peer review and any attached files.

Reviewer #1: **Yes: **Sebastian Baumgarten

Reviewer #2: No

---

## [Author Response · Author response to Decision Letter 0]

15 Jun 2022

Dear Prof Frischknecht

Thank you for the opportunity to revise this manuscript. Below is a list of revisions (red text) in response to the reviewers’ comments: we hope that the paper will now prove acceptable for publication.

Sincerely,

Catherine Merrick

Journal Requirements:

All checked 

This has been added in the online submission system

"UK Medical Research Council (grants MR/K000535/1 and MR/L008823/1) to CJM, Royal Society Kan Tong Po fellowship, Shenzhen Basic Research Project (JCYJ20180507181642811), Research Grants Council of the Hong Kong SAR, China Projects CityU 11101519, CityU 11100218, N_CityU110/17, CityU 21302317, Croucher Foundation Project No. 9509003, 9500030, State Key Laboratory of Marine Pollution Director Discretionary Fund, City University of Hong Kong projects 6000711, 7005503, 9680261, 9667222 to CKK. The funders had no role in study design, data collection and interpretation, or the decision to submit the work for publication."

We note that you have provided funding information that is not currently declared in your Funding Statement. However, funding information should not appear in the Funding section or other areas of your manuscript. We will only publish funding information present in the Funding Statement section of the online submission form. 

"The funders had no role in study design, data collection and analysis, decision to publish, or preparation of the manuscript." Please include your amended statements within your cover letter; we will change the online submission form on your behalf.

The cover letter now contains the correct funding statement, and it has been removed from the actual manuscript.

There are only 2 gels shown in this manuscript: both were already full-length (uncropped at top and bottom) and the one in supp Fig.1 was already full-width; the one in main Fig.1B has now been additionally provided as a full-width image in supp Fig .1. This is also noted in the cover letter. 

The ’data not shown’ have been added as a supplementary table (6) in the revised manuscript.

Done – reference formatting checked & one preprint updated to the published version.

Additional Editor Comments:

Dear Catherine,

thanks for sending your nice paper to Plos One. It has now been seen by two experts in the field, who, as you will see, liked the work but have some suggestions to improve the paper. I would encourage you to revise the manuscript according to their comments.

Sincerely

Freddy

Reviewer #1 

In this manuscript Dumetz et al. probe RNA secondary structures during the asexual replication of Plasmodium falciparum within human red blood cells. They use the well-established method Strucutre-seq that uses the chemical NAI to probe unpaired nucleotides and subsequent RT fall-off to interrogate RNA secondary structures on a transcriptome wide scale. Comparisons of their in vivo derived structurs to known, well characterized structures (e.g. tRNA) show the robustness of their method and I particularly like the ‘divergence’ score to categorize different RNA classes. Finally, the authors compare their RNA structure to translational efficiencies and RNA decay rates and find that some structures within RNA transcripts can influence those. Overall, the study is well designed, the methodology is clear, their findings well-presented and the study will overall lay the foundation to study dynamics of the RNA structurome across the P. falciparum life-cycle in the future.

Please find my comments to specific points below:

line 344ff: The structure-seq relies on RT fall-off. This also happens very frequently at sites of RNA modifications, especially in hypermodified molecules such as tRNA. For the tRNA where the authors do not find a ‘canonical’ structure, the RT fall-off because of modifications might make the probing of the actual structure impossible. This information should be added to the Discussion.

This valid point has been added (at line ~400)

What about different structures in different regions of mRNA transcripts, i.e. 5’UTR-CDS-3’UTR. UTRs in P. falciparum are substantially more A/U rich than the CDS. Please provide a paragraph/table/figure showing where different structures are located and if there is a potential bias.

This has now been discussed briefly at line ~370: ‘A second limitation of short-read sequencing was that it resulted in very limited coverage of untranslated regions, which in P. falciparum tend to be extremely A/U-biased (often over 90%), and also highly repetitive with many low-complexity regions. This precluded any comprehensive assessment of RNA structure in the UTRs versus CDSs of mRNAs.’

line 361: The authors do mention the previously published P. falciparum structuromes. How similar are the structures especially of mRNAs identified in the different studies?

Unfortunately, the previously published studies do not provide their reactivity data gene-by-gene - e.g. in dot-bracket format, which would facilitate the comparison of their paired/unpaired nucleotides with ours. (This is a young, developing field and there are not yet any community standards for publishing this type of data). Without this, it would be necessary to take their raw sequencing-read data and re-fold the entire transcriptome. Besides this being a very large task, the resultant comparison could anyway be somewhat flawed if they did not use exactly the same algorithms and parameters as us. Systematic comparisons between structuromes determined by different groups via different methods will likely be an important endeavour as this field develops; however, this is beyond the scope of our current paper.

line 325: It is not fully clear if all presented structures are derived from the NAI samples. If so, why did the authors chose to present only those in the main text although all samples where sequenced and present in the supplement? How different are the structures of the NAI and DMS samples?

This is already addressed (now at line 321): ‘Structural information was more comprehensive (and agreement between replicates was also better), for the NAI-probed datasets because all four unpaired nucleotides were detected. Subsequent analysis was therefore focused on the NAI dataset as the primary in vivo structurome’. We initially tried 2 different probing methods (DMS as well as NAI) because it was unclear whether both chemicals would efficiently access the Plasmodium RNA: this proved to be the case, and the information from the NAI datasets was therefore more complete. Because there is no well-validated way (to our knowledge) to blend data from DMS and NAI together, we chose to focus on the NAI datasets alone.

line 502: Different transcript regions have substantially different functions within a mRNA molecule. 5’UTRs are essential for translation and can even initiate protein synthesis cap-idenpendent with secondary structures, i.e. IRES. On the other hand 3’UTR are crucial for RNA decay. So how does the comparison with translational efficiencies and mRNA decay change when focusing only on structures on certain regions, i.e. 5’UTR and translation and 3’UTR and decay?

As per the point above: our coverage of UTRs, via basic Illumina sequencing, was too patchy to be confident in comparing this with CDSs. In future, some recently published improvements in RNAseq methodologies specifically to preserve A/U-rich Plasmodium sequences (e.g. ‘DAFT-seq’, Chappell et al 2020), could perhaps be used to ameliorate this problem.

Are there certain conserved structures in particular regions present in multiple different transcripts?

We have struggled with this question, which is rather general: i.e. how should we meaningfully define ‘particular regions’ of multiple transcripts? The simplest way to define regions is 5’UTR/CDS/3’UTR – but as noted above, our coverage of UTRs is much lower than of CDSs, precluding a meaningful analysis. We then considered seeking common RNA structures in the codes for common protein domains, but there are many hundreds of protein domains, versus only 4 broad categories of RNA structure – hairpin, stem, bulge and multi-stem-loop, so we did not find this to be meaningful either.

Please upload all raw and processed sequencing data to the SRA and GEO databases.

This is done, apologies for its omission from this manuscript (in a previous submission, it was provided in the metadata rather than the manuscript itself, but it has now been added to Materials & Methods after the section on sequencing). ‘Raw data (FASTQ) from sequencing for each sample is available from the European Nucleotide Archive (ENA) under accession PRJEB44384’.

Please ensure to accurately state “A/U” when talking about the transcriptome (e.g. in the title)

This has been checked/corrected throughout

Reviewer #2

Merrick and co-workers used a combination of chemical modification and RNA seq to inform on the secondary structure of mRNAs of the malaria-causing parasite Plasmodium falciparum. The authors were able to provide data on more than half of the parasite’s transcriptome and compare a AT-rich to a AT-balanced genome. The current study complements and expands insights from two recent studies on the same subject.

The presented data appear solid and the conclusions are scientifically sound. But some minor changes (for example Fig. 3) would help the reader to better access the significance of this work. Below are some specific minor comments, which should help to further strengthen this manuscript.

Line 214: Technically the parasite does not live “entirely” inside a host cell, many stages are extracellularly. Please revise.

This has been corrected

Sentence starting in line 250 may be clearer if stating that nucleotides were unpaired before labelling reaction.

This has been corrected

The labels of Suppl. Fig. 3 are very small.

This has been corrected

For Fig. 2C, would it be possible to have the experimentally-determined structures superimposed on the “canonical” folding? This would strengthen the point. While Fig. 2C and Suppl. Fig. 4 clearly show the strength of the approach in detecting the expected canonical structures of some tRNAs, showing some of the non-canonical structures in the main figure would be appreciated.

In figure 2C, it is the experimentally-determined structures that are shown. In the original figure, we showed only experimentally-determined structures that were canonical (non-canonical versions were in the supp figure). As requested, we have now shown a non-canonical example as well, for direct side-by-side comparison within the main figure. 

Revise formatting of the reference in line 358.

All references are checked/updated in the revision.

End of the sentence in line 403: . instead of ,

This has been corrected.

Fig. 3, please provide a scale for the different shades of red/fold enrichment. Which P-value cut-off was used to identify significantly enriched GO terms? Also, the figure would be easier to understand, if panels A, B, and C had labels indicating the divergence tiers. It is unclear how the numbers of enriched GO terms in the text correspond to the number of GO terms shown in the figure, please explain.

We appreciate the reviewer’s pointing out that Figure 3 (a representation of GO terms analysis) was not fully quantitative. We have therefore re-done the analysis, changing the representation to a scatter-plot format, which still represents the relatedness between similar GO terms, but is also quantitative both in terms of enrichment (see rainbow scale) and number of GO terms (see size of circles). We have also labelled the 3 plots with Tiers 1-3. 

The Fig. 4A would benefit from a legend describing what the green/red shading and the boxes indicate. Please also explain here what the dot/bracket sequences mean.

This information is all in the legend, but we appreciate the reviewer’s pointing out that the figure is complex to look at without visual legend as well. A visual legend has been added to a revised figure 4.

Is the term “significant” (line 435) used to indicate statistical significance? If so, please provide information on the tests. If not, please rephrase.

This has been corrected to ‘major’

Fig. 5 and 6 caption: It is not the “life cycle” but the development inside erythrocytes.

This has been corrected

While the other two studies investigating the structure of P. falciparum mRNAs are mentioned, this manuscript would benefit from an extended discussion of the results from the other studies.

We do already have within our discussion several paragraphs of comparative insight concerning the two other existing studies: 

a) lines 585-594 comparing the overall coverage of our structurome vz that of Qi et al., 

b) lines 636-647 comparing our findings about RNA structure vz translational-efficiency,

c) lines 685-690 concerning the effect of RNA binding proteins, as suggested by Qi et al. 

It is difficult to provide much more insightful discussion, particularly of Alvarez et al., since these authors used an entirely different methodology.

---

## [Editor Report · Decision Letter 1]

20 Jun 2022

The in vivo RNA structurome of the malaria parasite Plasmodium falciparum, a protozoan with an A/U-rich transcriptome

PONE-D-22-07594R1

Dear Dr. Merrick,

We’re pleased to inform you that your manuscript has been judged scientifically suitable for publication and will be formally accepted for publication once it meets all outstanding technical requirements.

Kind regards,

Friedrich Frischknecht

Academic Editor

PLOS ONE

Additional Editor Comments (optional):

thanks for providing a constructive revision, in my view the paper can now be accepted. I wonder only if in the table 2 all numbers could be limited to two digits after the . ?
---

## [Editor Report · Acceptance letter]

16 Aug 2022

PONE-D-22-07594R1 

The *in vivo* RNA structurome of the malaria parasite *Plasmodium falciparum*, a protozoan with an A/U-rich transcriptome  

Dear Dr. Merrick:

I'm pleased to inform you that your manuscript has been deemed suitable for publication in PLOS ONE. Congratulations! Your manuscript is now with our production department. 

Kind regards, 

on behalf of

Prof. Dr. Friedrich Frischknecht 

Academic Editor

PLOS ONE